# Clinicopathological Features and Prognosis of Lung Adenocarcinoma Presenting as Ground-Glass Opacity: A Single-Center Experience

**DOI:** 10.3390/cancers17183016

**Published:** 2025-09-16

**Authors:** Monica Casiraghi, Lara Girelli, Attilio Elettore, Luca Bertolaccini, Antonio Mazzella, Claudia Bardoni, Mariano Lombardi, Valeria Midolo, Giuseppe Petralia, Patrick Maisonneuve, Juliana Guarize, Lorenzo Spaggiari

**Affiliations:** 1Department of Thoracic Surgery, IEO-European Institute of Oncology IRCCS, 20141 Milan, Italy; lara.girelli@ieo.it (L.G.); attilio.elettore@studenti.unimi.it (A.E.); luca.bertolaccini@gmail.com (L.B.); antonio.mazzella@ieo.it (A.M.); claudia.bardoni@ieo.it (C.B.); juliana.guarize@ieo.it (J.G.); lorenzo.spaggiari@ieo.it (L.S.); 2Department of Oncology and Hemato-Oncology, University of Milan, 20122 Milan, Italy; giuseppe.petralia@ieo.it; 3Department of Pathology, IEO-European Institute of Oncology IRCCS, 20141 Milan, Italy; mariano.lombardi@ieo.it (M.L.); valeria.midolodeluca@ieo.it (V.M.); 4Department of Radiology, IEO-European Institute of Oncology IRCCS, 20141 Milan, Italy; 5Division of Epidemiology and Biostatistics, IEO-European Institute of Oncology IRCCS, 20141 Milan, Italy; patrick.maisonneuve@ieo.it

**Keywords:** adenocarcinoma, ground-glass opacities, lung surgery, lung cancer

## Abstract

This single-center study aimed to explore the relationship between radiological appearance and pathological findings in lung adenocarcinoma presenting as ground-glass opacity (GGO) and their impact on lung cancer-specific survival (LCSS) and recurrence. A total of 172 patients underwent lung resection for adenocarcinoma, classified into pure GGO (consolidation-to-tumor ratio (CTR) 0) and partial solid GGO (CTR > 0 to 1.0) based on CT imaging. Invasive adenocarcinoma was significantly more common in partial solid GGO cases. Histologically, higher-grade tumors (G2 and G3) correlated with increasing CTR. Disease recurrence occurred in 12.2% of patients, more frequently among those with invasive histology and partial solid lesions. Multivariate analysis showed that age over 60, clinical stage ≥ cIB, and sublobar resection were significantly associated with relapse. The study concludes that pure GGO lesions are typically less invasive and have better prognoses, but recurrence risk increases with more advanced clinical features and limited surgical approaches, underscoring the need for tailored surgical planning.

## 1. Introduction

To date, different studies have already reported excellent prognosis in patients undergoing radical surgery for primary lung cancer radiologically presented as GGO. However, it becomes essential to understand how much this radiological presentation corresponds to lower tumor invasiveness from a pathological perspective and, consequently, to a better prognosis.

The term “ground-glass opacity” (GGO) refers to an area of increased attenuation of the lung parenchyma visible on CT scans, with preservation of the bronchial and vascular structures [1].

GGO images could correspond to different pathological conditions from benign disease to preinvasive lesions such as AAH and AIS, but also malignant tumors such as MIA and IA with a predominant lepidic pattern [2]. However, while AAH typically presents as pure GGO smaller than 5 mm, AIS usually appears as a GGO but with larger dimensions and greater density, and in most cases of MIA, a partially solid component begins to appear until the GGO component becomes progressively less presented into the IAs. Several studies have already shown that the radiological GGO component corresponds, in fact, to the lepidic histological element, while the solid part is usually the invasive one [3,4]. However, these correlations are not usually only supposed, and a definitive diagnosis can only be made with a histological analysis [5].

As many researchers have reported, the presence of a GGO component in lung nodules and its proportion, represented as the CTR and calculated by dividing the maximum solid component diameter by the maximum tumor diameter, could strongly correlate with patients’ prognoses and pathological findings, even for those with the same clinical stage of disease [6,7,8,9,10,11].

This study aimed to retrospectively analyze data obtained from patients undergoing pulmonary resection for adenocarcinoma radiologically presented as a GGO lesion. It focused on the impact of the GGO component on pathological-anatomical findings, survival, and potential disease recurrence.

## 2. Materials and Methods

We retrospectively analyzed the medical records of 950 consecutive patients who underwent lung resection for lung adenocarcinoma at the Division of Thoracic Surgery at the European Institute of Oncology in Milan between 2006 and 2022. The study was conducted in accordance with the Declaration of Helsinki and approved by the Institutional Review Board of European Institute of Oncology (protocol code 4462 and approved on 19 December 2023).

Out of 950 patients, 172 (5.5%) patients were selected for the study based on the following inclusion criteria: (1) presentation at chest CT scan of a single pulmonary nodule that was either pure GGO (CTR 0) or GGO with a solid component (CTR ≤ 0.5, 0.5 < CTR ≤ 0.75, and 0.75 < CTR ≤ 1.0) (Figure 1); (2) surgical resection of the lesion via wedge resection, anatomical segmentectomy, or lobectomy; and (3) final histological diagnosis of adenocarcinoma.

### 2.1. Pre-Operative Evaluation

Since 2006, our preoperative staging routinely included a total body CT scan and PET-FDG [12]. All nodules were measured following the Fleischer Society recommendation: chest CT scans were performed with 1 mm sections, and the maximum and minimum diameters of the nodules were measured to derive the average diameter. The solid component was measured using a mediastinal window [13].

Thus, an experienced radiologist reviewed all radiological CT scan images to determine the tumor size, the size of the solid component, and the size of the GGO component, and thus the CTR. Lesions were categorized into four groups: pure GGO with CTR = 0, GGO-predominant nodules with CTR ≤ 0.5, solid-predominant nodules with CTR between 0.5 and 0.75, and solid-predominant nodules with CTR between 0.75 and 1.

Patients were all stage-based on the 8th TNM AJCC staging system [14]; patients staged in the past with different TNM editions were restaged based on the recent one (8th TNM ed.).

### 2.2. Surgery

Surgical procedures were performed either via lateral muscle-sparing thoracotomy or minimally invasive techniques such as robotic-assisted thoracic surgery (RATS) or video-assisted thoracic surgery (VATS). Indication for the different surgical approaches was based only on surgeons’ and patients’ preferences, and it did not change over the years considering that we started performing RATS and VATS in 2006. Lymphadenectomy was performed according to the American Thoracic Society classification, removing all lymphatic tissue from stations 2R, 4R, 7, and 10R for right-sided tumors and stations 5, 6, 7, and 10 L for left-sided tumors.

### 2.3. Histopathological Examination

Two experienced pathologists analyzed the slides, confirming the diagnosis of adenocarcinoma, defining the invasiveness of the lesion (in situ, minimally invasive, invasive), and grading it (G1, G2, G3) according to standardized criteria following the 2015 World Health Organization (WHO) [14]. In cases with doubtful aspects, an immunohistochemical panel consistent with CK7, TTF-1, and p40 was performed to confirm the invasive component.

### 2.4. Post-Operative Follow-Up

Post-operative follow-up included a physical exam and chest and upper abdomen CT scans every 6 months for the first 2 years and every 12 months for the next 3 years.

Recurrence at the surgical site (hilar/mediastinal region or lung parenchyma near the previous resection) was classified as local, whereas relapses into the thoracic cavity (ipsilateral and contralateral, such as new pulmonary nodules) were considered regional; extra thoracic lesions were classified as distant recurrences.

### 2.5. Statistical Analysis

Differences between patient groups were assessed using the chi-square test for categorical variables, the Wilcoxon rank test for continuous variables, or the Mantel–Haenszel test for ordinal variable trends.

Overall survival (OS) and cancer-specific survival (CSS) were estimated using the Kalbfleisch and Prentice method, accounting for competing events, and the Gray test was used to assess survival differences between the two groups. The cumulative incidence of recurrence was evaluated at 5- and 10-year follow-ups. Univariable and multivariable Cox proportional hazards regression was used to identify factors associated with the development of relapse, employing the Fine and Gray sub-distribution hazard model to account for competing risks. Statistical analyses were performed using SAS software (version 9.4, SAS Institute, Cary, NC, USA). Statistical significance was defined as *p* < 0.05.

## 3. Results

One-hundred and seventy-two patients underwent lung resection for an adenocarcinoma that radiologically appeared as pure or partially solid GGO.

Patients’ characteristics, surgical details, and anatomopathological tumor features are included in Table 1.

At radiological evaluation, 90 (52.3%) were pure GGO, while 82 (47.7%) were partial solid GGO with a solid component. Among those with a solid component, 56 (32.6%) had CTR ≤ 0.5, 18 (10.5%) patients had a CTR between 0.5 and 0.75, and 8 (4.7%) had a CTR between 0.75 and 1. FDG PET was negative on lung lesions in 93 patients (54.1%). Thirty-eight (22.1%) patients underwent surgery using standard thoracotomy, 48 (27.9%) underwent surgery via VATS, and 86 (50.0%) underwent surgery with the RATS approach. The most performed surgical resection was lobectomy (n = 130; 75.6%), followed by wedge resection (n = 32; 18.6%) and anatomical segmentectomy (n = 10; 5.8%). Hilar-mediastinal lymphadenectomy was performed in 145 (84.3%) patients, including all 130 (100%) lobectomies, 10 (100%) anatomical segmentectomies, and 5 (15.6%) wedge resections. Oncological radicality was achieved in all patients (100%).

Most lesions were found to be IA (n = 136; 79.1%), while the remaining tumors were MIA (n = 31; 18.0%) and AIS (n = 5; 2.9%). The most frequent subtype of adenocarcinoma was the lepidic predominant pattern (n = 126; 73.2%), which was mixed with the acinar and papillary type in 86 cases (50%). The median tumor size was 13 mm (range: 3–90 mm). Tumor grading was well-differentiated (G1) in 90 (52.3%) cases, moderately differentiated (G2) in 72 (41.9%) cases, and poorly differentiated in 10 (5.8%) cases.

Regarding pT classification, 69 (40.1%) patients were classified as pT1a, with 80 (46.5%) as pT1b, 16 (9.3%) as pT1c, 4 (2.3%) as pT2a, and 2 (1.2%) as pT2b. One (0.6%) case involved an adenocarcinoma with a 90 mm invasive component, classified as pT4. The predominant pathological stage was stage IA (n = 163; 94.8%); three patients were classified as stage IB (1.7%), with two as stage IIA (1.2%), three as stage IIB (1.7%), and one as stage IIIA (0.6%). Lymph node metastases were found in only two (1.2%) cases; both of them were nodule with a radiologic solid component (between 0.75 and 1) and invasive adenocarcinomas.

### 3.1. Univariate and Multivariate Analysis

In the univariate analysis (Table 2), recurrence was significantly associated with age over 60 (*p* = 0.025), a clinical stage higher than cIA (*p* = 0.010), sublobar resections (*p* = 0.035), a pathological T stage higher than T1a (*p* = 0.0001), and a pathological stage higher than IA (*p* < 0.0001).

From the multivariate analysis (Table 3), it is seen that patients over 60 years had a risk of recurrence 8.92 times higher (*p* = 0.037) compared to patients under 60 years old, and patients undergoing sublobar resection had a 3.17 times higher risk of recurrence (*p* = 0.005) compared to those who underwent lobectomy. In addition, patients with clinical stage IB or higher had a 5.06 times higher risk of recurrence (*p* = 0.002) compared to patients with clinical stage IA.

### 3.2. Prognostic Factors and Survival

OS, CSS, and cumulative incidence of relapses of the 172 patients are shown in Figure 2. OS and CSS were not statistically different between the pure GGO group and the partial solid GGO group, confirming the good prognosis related to the presence of GGO component. Only two (1.2%) patients died of the disease, and both of them had CTR values between 0.75 and 1.

Twenty-one (12.2%) patients experienced disease recurrence: 4 (2.3%) patients had local recurrence, in particular in the same lobe, and all of them had sublobar resection (2 with segmentectomy and the other 2 with wedge resection), 16 (9.3%) had regional recurrence (all of that had new pulmonary nodules), and 1 (0.6%) had distant recurrence. In patients with recurrence, 16 (9.3%) were IA, and 5 (2.9%) were MIA; radiologically, 13 (7.5%) were pure GGO, and 8 (4.6%) cases had a solid component.

Incidence of recurrence based on age, type of surgery, and stage are shown in Figure 3A–C, respectively.

Comparing the group with pure GGOs and the groups with solid components, IA was significantly more frequent (*p* = 0.0006) in the group with solid nodules, and AIS was found only in the pure GGO group, showing a relation between CTR and the anatomopathological characteristics of the tumor. In addition, in the pure GGO group, most tumors were well-differentiated G1. In contrast, in the solid component nodules group, the majority of tumors were moderately differentiated G2, and the percentage of poorly differentiated G3 tumors increased together with the CTR increasing (*p* = 0.0002) (Table 4).

The cumulative incidence of relapses was not statistically significant between pure GGO and partial solid GGO (Appendix A), whereas a significant increase of relapses was evident in patients with pure GGO undergoing sublobar resection (*p* = 0.01) (Appendix A).

## 4. Discussion

Literature largely agrees that the CTR is a reliable radiological indicator of tumor invasiveness in lung adenocarcinomas. Higher CTR values, which reflect a greater proportion of solid components relative to the total tumor size on CT scan, have been consistently associated with increased histopathological invasiveness and worse prognosis [1,2,3], as has already been demonstrated by different studies validating the 8th TNM Edition classification, which have shown a strong correlation between the solid size measured on thin-section CT and the invasive size found in pathological specimens [4,5,6]. GGO, inversely related to CTR, is often a favorable prognostic factor, highlighting the importance of CTR in surgical decision-making and outcome prediction [7,8].

Prospective studies such as the Japan Clinical Oncology Group 0201 trial have further supported the use of CTR and consolidation size as non-invasive predictors of pathological invasiveness, guiding tailored treatment strategies for early-stage lung cancer patients [9,10]. Suzuki et al. showed how a pathological non-invasiveness of lesions was related to a CTR < 0.5 and also that when using a CTR cut-off of 0.25, the specificity and sensitivity predicting tumor non-invasiveness were 98.7% and 16.2%, respectively [9]. In a systematic review published by Gao et al. in 2017, the authors reported that non-mucinous adenocarcinoma in situ usually had a radiological finding on CT as a pure GGO, minimally invasive adenocarcinoma appears as a partially solid nodule with a CTR < 0.5 and a solid component <5 mm, while invasive adenocarcinoma appears as a partially solid nodule with a solid component >5 mm and a CTR > 0.5 [2]. Even Fu and colleagues showed that increasing the CTR increased the chances of finding poorly differentiated tumors (mainly micropapillary or solid adenocarcinomas) [15].

Also in our case series, we found a relation between lesion invasiveness and the presence of a solid component (*p* < 0.0007); adenocarcinomas in situ appeared on CT as pure GGO, while nodules with a solid component were more frequently invasive, as also reported in the review by Gao and colleagues [2]. Our result confirmed the JCOG0201 study, as neither study demonstrated non-invasiveness with a CTR cut-off of 0.5, suggesting that a lower cut-off should be used to predict tumor non-invasiveness [10]. In another study published by Aokage, no lesions with a CTR < 0.5 were shown to be predominantly solid-pattern (poorly differentiated) adenocarcinomas. At the same time, these constituted about 10% of the lesions with a CTR > 0.5 [4].

Similarly, we demonstrated that a solid component of CT and an increase in CTR were associated with increased tumor aggressiveness (*p* < 0.0001), even if it is not always clear whether CTR was an independent prognostic factor. Instead, the clinical stage and the size of the solid component were significant, confirming the literature results [16,17,18].

Several studies argued that the proportion of different components of the nodule was not a relevant prognostic factor. However, the mere presence or absence of the GGO component was usually related to a different prognosis, suggesting different biological behavior. In these studies, the size of the solid element impacted the patient’s prognosis only in the group with entirely solid nodules. In contrast, in the group with partially solid nodules, the CTR and the size of the solid part, even if they reflected tumor invasiveness, did not impact overall survival or disease-free survival [16,17,18], confirming that the presence of a GGO component reflects less aggressive biological behavior of the disease, predominantly lepidic growth tumors [17,18].

In 2018, Suzuki et al., studying tumors with GGO > 3 cm, divided them based on CTR and showed that nodules with CTR less than 0.5 did not have any lymph node metastasis, with excellent 5-year overall survival up to 100%, which was different from patients with a CTR > 0.5, who had a significantly worse survival rate [19]. The same results were confirmed in 2020 by Wang et al., who showed how solid component ratios were possible independent risk factors for lymph node metastasis in patients with stage IA lung adenocarcinoma, whereas they were rare in patients with pure GGO or GGO-dominant lung adenocarcinoma [20]. Even in our previous study, published in 2010, we analyzed the behavior of 219 pathological NSCLCs to identify criteria predictive of nodal involvement and the role of cancer size in lymph node metastases; despite all nodal metastases occurring among patients with sizes larger than 10 mm and SUVs higher than 2.0 with a 22% rate of nodal involvement (*p* < 0.0001), none of the patients with non-solid tumors had lymph node involvement [21].

Also in our study, only 1.2% of tumors showed lymph node metastasis into the final pathological analysis, even if lymphadenectomy was not performed in all patients but in 84% of the cases. However, lymphadenectomy was not a significant prognostic factor, and none of the patients had lymph node relapses.

Hattori reported that the 5-year OS for clinical stage IA adenocarcinoma was 91.2% for the group showing a radiological GGO component and only 68.9% for the group with entirely solid nodules, highlighting how even the radiological appearance was correlated with greater tumor aggressiveness and an increased risk of recurrence with lower survival [22,23].

In our study, only two (1.1%) patients died with disease, confirming the excellent prognosis for adenocarcinomas presenting as lesions with a GGO component on CT, regardless of disease stage or CTR.

Analyzing literature data based on surgical type of resection, Ye et al. showed that 5-year OS and RFS, in adenocarcinomas nodules with GGO ≤ 2 cm, were similar between sublobar resections and lobectomy [24].

The surgical management of GGO-associated lung adenocarcinomas remains a critical topic of debate, particularly regarding the choice between lobar and sublobar resection. GGOs often represent less invasive tumors with favorable prognoses, raising the question of whether less extensive surgery can achieve oncologic outcomes comparable to standard lobectomy while preserving lung function. Recent evidence, including results from pivotal trials such as the JCOG0802 study, has provided significant insight into this issue [25]. The JCOG0802 trial demonstrated that segmentectomy achieved non-inferior overall survival compared to lobectomy for small peripheral non-small cell lung cancers, including those presenting with GGO components, showing a 5-year RFS of 88% for segmentectomy compared to 87.9% for lobectomy, respectively, even if with local recurrences significantly more frequent in the segmentectomy group (10.5% vs. 5.4%). This has shifted the surgical paradigm toward more conservative resections in select patients with early-stage adenocarcinomas, balancing oncologic safety with functional preservation. However, appropriate patient selection remains paramount, considering tumor size, CTR, and invasive potential to optimize outcomes and minimize recurrence risks.

In addition, the CALGB140503 trial, which analyzed a series of patients with stage T1aN0 NSCLC (cT < 2 cm) undergoing lobectomy or sublobar resection (wedge or segmentectomy), showed that RFS and OS were similar between the two groups, demonstrating the non-inferiority of sublobar resections [26].

Although our study did not show differences in OS between sublobar resections and lobectomy in any clinical stage, it showed higher incidence of relapses for sublobar resections, even if not statistically significant, as shown in the literature [25,26,27,28,29,30]. Even if this was not the aim of our study (comparing sublobar and lobar resection), this finding showed that even in the case of GGOs or partial solid lesions, surgeons need to be aware of possible relapses in the case of sublobar resection and always rigorously follow rules and guidelines for sublobar resections [25,26,27,28,29,30]

The retrospective nature of this study definitely limits and influences the variables included in the analysis, as well as the “single arm” analysis with no comparison with multiple GGOs or non-surgical patients, limiting our results and conclusions; in addition, the mono-center experience could be considered a bias mainly due to the patient’s selection but is also a point of strength, accounting for minor variability in terms of oncological experience compared to the multi-center studies.

## 5. Conclusions

In conclusion, this study highlighted lower invasiveness in tumors radiologically presenting as pure GGOs compared to those with a solid component, with increased aggressiveness directly proportional to the percentage of the solid component. However, prognosis was not found to be correlated with CTR but with the GGO presence itself; instead, it was associated with tumor stage and type of surgical resection performed. Nevertheless, randomized studies will be necessary to confirm these conclusions and provide further evidence supporting therapeutic strategies for lung cancers with mixed or GGO-predominant components.

## Figures and Tables

**Figure 1 cancers-17-03016-f001:**
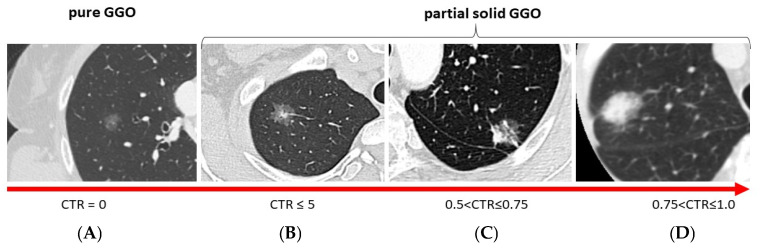
Presentation at chest CT scan of a single pulmonary nodule: (**A**) pure GGO (CTR 0); (**B**) GGO with a solid component CTR ≤ 0.5, (**C**) GGO with a solid component 0.5 < CTR ≤ 0.75, and (**D**) GGO with a solid component 0.75 < CTR ≤ 1.0.

**Figure 2 cancers-17-03016-f002:**
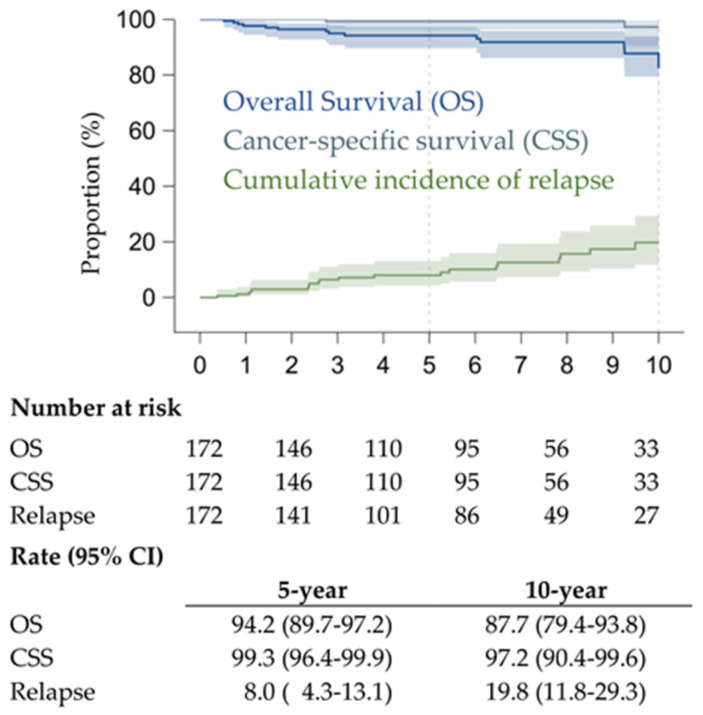
Overall survival, cancer-specific survival, and cumulative incidence of relapses.

**Figure 3 cancers-17-03016-f003:**
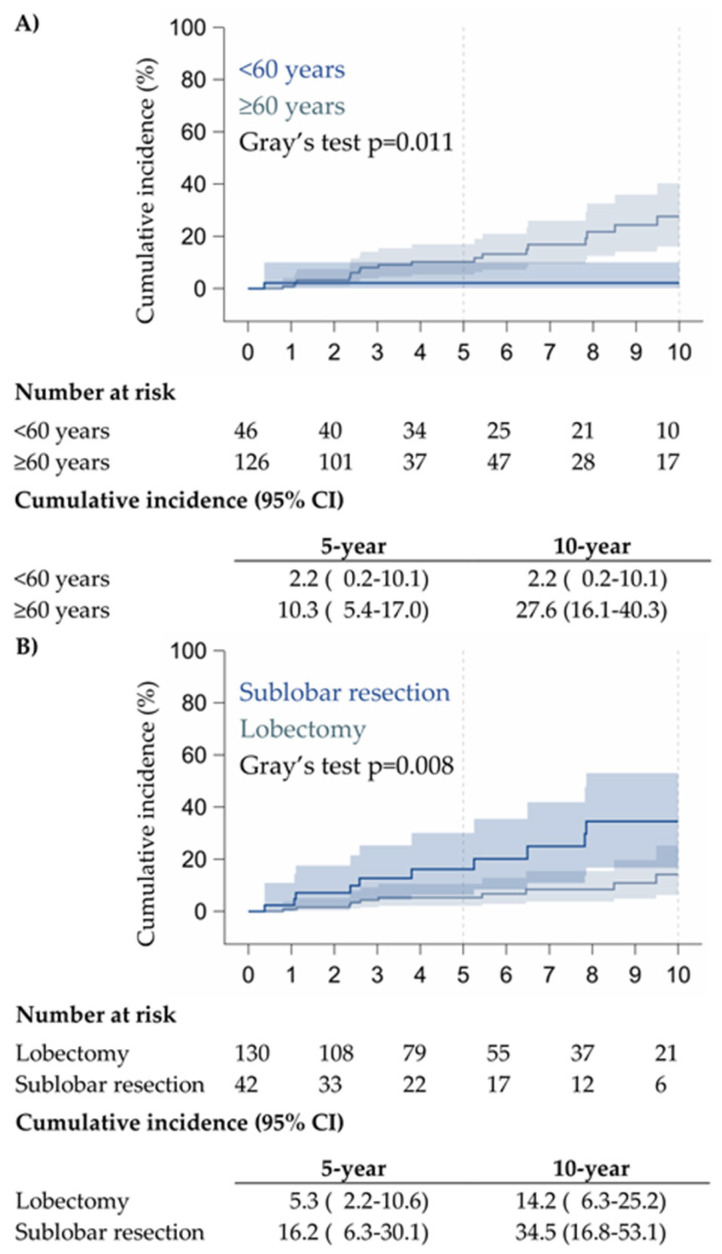
Cumulative incidence of relapses in patients with GGO according to age (**A**), surgery (**B**), and clinical stage (**C**).

**Table 1 cancers-17-03016-t001:** Patients’ clinical and surgical characteristics.

	Patients		Patients
	N (%)		N (%)
**All patients**	172 (100.)	**Histological subtype**	
		Acinar	32 (18.6)
**Sex**		Papillary	7 (4.1)
Male	74 (43.0)	Lepidic predominant	41 (23.8)
Female	98 (57.0)	Mixed Acinar + papillary	6 (3.5)
**Age**		Acinar + lepidic predominant	70 (40.7)
Median [range]	65 [38–81]	Papillary + lepidic predominant	16 (9.3)
<60	46 (26.7)	**CTR**	
60–69	69 (40.1)	Pure GGO	90 (52.3)
70+	57 (33.1)	≤0.50	56 (32.6)
**Clinical stage**		0.50–0.75	18 (10.5)
IA	166 (96.5)	0.75–1.00	8 (4.7)
IB/II/III	6 (3.5)	**Invasiveness**	
**Site**		AIS	5 (2.9)
RUL	74 (43.0)	MIA	31 (18.0)
ML	11 (6.4)	IA	136 (79.1)
RLL	28 (16.3)	**Size**	
LUL	38 (22.1)	Median [range]	13 [3–90]
LLL	21 (12.2)	≤10	61 (35.5)
**Access**		11–20	87 (50.6)
Open	38 (22.1)	>20	24 (14.0)
Robot	86 (51.7)	**pT**	
VATS	48 (27.9)	1a	69 (40.1)
**Surgery**		1b	80 (46.5)
Lobectomy	130 (75.6)	1c	16 (9.3)
Segmentectomy	10 (5.8)	2a	4 (2.3)
Wedge	32 (18.6)	2b	2 (1.2)
**Lymphadenectomy**		3–4 *	1 (0.6)
No	27 (15.7)	**pN**	
Yes	45 (26.2)	pN−	143 (83.1)
**Radicality**		pN+	2 (1.2)
Yes	172 (100.)	pNx	27 (15.7)
No	-	**Stage**	
**PET**		IA	163 (94.8)
Missing	6 (3.5)	IB	3 (1.7)
Negative	93 (54.1)	IIA	2 (1.2)
Positive	73 (42.4)	IIB	3 (1.7)
**Radiologic GGO**		IIIA	1 (0.6)
Pure GGO	90 (52.3)	**Grading**	
Solid component	82 (47.7)	G1	90 (52.3)
**CTR**		G2	72 (41.9)
Pure GGO	90 (52.3)	G3	10 (5.8)
≤0.50	56 (32.6)	**Post-operative treatment**	
0.50–0.75	18 (10.5)	No	170 (98.8)
0.75–1.00	8 (4.7)	Yes	2 (1.2)

*: One (0.6%) case involved an adenocarcinoma with a 90 mm invasive component, classified as pT4. MIA: minimally invasive adenocarcinoma; IA: invasive adenocarcinoma; AIS: adenocarcinoma in situ.

**Table 2 cancers-17-03016-t002:** Univariate analysis for relapse-free survival.

	Patients (n = 172)	Relapses (n = 21)	Grays’ *p*-Value	Univariate HR (95% CI)
**Age**				
<60	46	1		1.00
60–69	69	11	**0.025**	**8.35 (1.04–67.2)**
70+	57	9		**10.2 (1.23–84.7)**
**Clinical stage**				
IA	166	18		1.00
IB/II/III	6	3	**0.010**	**4.94 (2.07–11.8)**
**Surgery**				
Lobectomy	130	11		1.00
Segmentectomy	10	4	**0.035**	**3.99 (1.32–12.1)**
Wedge	32	6		2.48 (0.92–6.71)
**Lymphadenectomy**				
No	27	5		1.00
Yes	145	16	0.20	0.52 (0.18–1.48)
**PET**				
Negative	93	11		1.00
Positive	73	8	0.17	1.18 (0.48–2.87)
Missing	6	2		4.29 (0.87–21.1)
**GGO**				
Pure GGO	90	13		1.00
Solid component	82	8	0.57	0.77 (0.32–1.83)
**CTR**				
Pure GGO	90	13		1.00
≤0.50	56	5		0.72 (0.26–1.98)
0.50–0.75	18	2		0.95 (0.21–4.38)
0.75–1.00	8	1	0.94	0.74 (0.10–5.82)
**Histological subtype**				
Acinar	32	4		1.00
Papillary	7	0		-
Lepidic predominant	41	4		0.82 (0.21–3.26)
Mix Acinar + papillary	6	0		-
Acinar + lepidic predominant	70	11		1.33 (0.42–4.21)
Papillary + lepidic predominant	16	2	0.73	0.86 (0.17–4.38)
**Invasiveness**				
AIS	5	0		-
MIA	31	5		1.15 (0.44–3.00)
IA	136	16	0.69	1.00
**Size**				
≤10	61	7		1.00
11–20	87	11		1.43 (0.55–3.72)
>20	24	3	0.67	1.55 (0.41–5.88)
**pT**				
pT1	165	19		1.00
pT2	6	1		1.61 (0.41–6.34)
pT3–4 *	1	1	**0.0001**	**22.6 (10.7–47.8)**
**pN**				
pN−	143	16		1.00
pN+	2	0		-
pNx	27	5	0.38	1.89 (0.66–5.41)
**Stage**				
IA/IB	166	18		1.00
IIA/B	5	2		2.58 (0.86–7.75)
IIIA	1	1	**<0.0001**	**23.3 (10.9–49.6)**
**Grading**				
G1	90	11		1.00
G2	72	7		1.00 (0.40–2.51)
G3	10	3	0.38	2.46 (0.64–9.44)

*: One (0.6%) case involved an adenocarcinoma with a 90 mm invasive component, classified as pT4. MIA: minimally invasive adenocarcinoma; IA: invasive adenocarcinoma; AIS: adenocarcinoma in situ. *p*-value in bold indicates a statistically significant difference with a *p*-value less than 0.05.

**Table 3 cancers-17-03016-t003:** Multivariate analysis for relapse-free survival.

	HR (95% CI) *	*p*-Value	Time-Dependent Area Under the Curve (95% CI)
**Age**			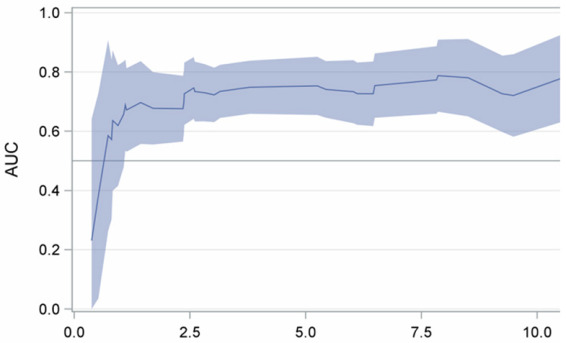
<60 years	1.00		
≥60 years	8.92 (1.14–69.9)	0.037	
**Type of surgery**			
Lobectomy	1.00		
Sublobar resection	3.17 (1.42–7.10)	**0.005**	
**Clinical stage**			
IA	1.00		
IB or more	5.06 (1.80–14.2)	**0.002**	

* stratified for gender. *p*-value in bold indicates a statistically significant difference with a *p*-value less than 0.05.

**Table 4 cancers-17-03016-t004:** Association between selected characteristics and CTR in patients with GGO.

	All	Consolidation-to-Tumor Ratio (CTR)				
	Patients	Pure GGO	≤0.50	0.50–0.75	0.75–1.00	
	N (%)	N (%)	N (%)	N (%)	N (%)	*p*-Value *
**All patients**	**172 (100.)**	**90 (100.)**	**56 (100.)**	**18 (100.)**	**8 (100.)**	
**Invasiveness**						
AIS	5 (2.9)	5 (5.6)	0 (0.0)	0 (0.0)	0 (0.0)	
MIA	31 (18.0)	24 (26.7)	6 (10.7)	0 (0.0)	1 (12.5)	
IA	136 (79.1)	61 (67.8)	50 (89.3)	18 (100.)	7 (87.5)	**0.0006**
**Size**						
≤10	61 (35.5)	39 (43.3)	18 (32.1)	1 (5.6)	3 (37.5)	
11–20	87 (50.6)	43 (47.8)	30 (53.6)	10 (55.6)	4 (50.0)	
>20	24 (14.0)	8 (8.9)	8 (14.3)	7 (38.9)	1 (12.5)	**0.004**
**pT**						
1a	69 (40.1)	44 (48.9)	20 (35.7)	1 (5.6)	4 (50.0)	
1b	80 (46.5)	39 (43.3)	28 (50.0)	10 (55.6)	3 (37.5)	
1c	16 (9.3)	4 (4.4)	5 (8.9)	6 (33.3)	1 (12.5)	
2a	4 (2.3)	2 (2.2)	1 (1.8)	1 (5.6)	0 (0.0)	
2b	2 (1.2)	1 (1.1)	1 (1.8)	0 (0.0)	0 (0.0)	
3–4 *	1 (0.6)	0 (0.0)	1 (1.8)	0 (0.0)	0 (0.0)	**0.020**
**Stage**						
IA	163 (94.8)	85 (94.4)	53 (94.6)	17 (94.4)	8 (100.)	
IB	3 (1.7)	1 (1.1)	1 (1.8)	1 (5.6)	0 (0.0)	
IIA	2 (1.2)	1 (1.1)	1 (1.8)	0 (0.0)	0 (0.0)	
IIB	3 (1.7)	3 (3.3)	0 (0.0)	0 (0.0)	0 (0.0)	
IIIA	1 (0.6)	0 (0.0)	1 (1.8)	0 (0.0)	0 (0.0)	0.46
**Grading**						
G1	90 (52.3)	61 (67.8)	23 (41.1)	3 (16.7)	3 (37.5)	
G2	72 (41.9)	24 (26.7)	31 (55.4)	13 (72.2)	4 (50.0)	
G3	10 (5.8)	5 (5.6)	2 (3.6)	2 (11.1)	1 (12.5)	**0.0002**

*: One (0.6%) case involved an adenocarcinoma with a 90 mm invasive component, classified as pT4. MIA: minimally invasive adenocarcinoma; IA: invasive adenocarcinoma; AIS: adenocarcinoma in situ. *p*-value in bold indicates a statistically significant difference with a *p*-value less than 0.05.

## Data Availability

The original contributions presented in this study are included in the article.

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
