# Peer review of "Clinicopathological Features and Prognosis of Lung Adenocarcinoma Presenting as Ground-Glass Opacity: A Single-Center Experience"

_cancers, 2025, doi:10.3390/cancers17183016_

Round 1

Reviewer 1 Report

Comments and Suggestions for Authors

Major revision

1. It is necessary to include the original pathohistological diagnosis and the type of adenocarcinoma in solid component in Tables of Results 

2. For the sake of better illustration, the radiological presentation should include  figures of  characteristic CTR on CT scans with corresponding histological slides. 

Minor revisions

22 In Abstract CTR explain what means

33 lepidic growth of tumor cells 

65 less significant ...maybe ... smaller  or less presented

Author Response

Question 1:  It is necessary to include the original pathohistological diagnosis and the type of adenocarcinoma in solid component in Tables of Results 

Answer 1: Pathohistological diagnosis and the type of adenocarcinoma in solid component details were added to table 1 and 2 all the details. We also added a paragraph into the result section at page 5, line 168 “The most frequent subtype of adenocarcinoma was the lepidic predominant pattern (n=126;73.2%), which was mixed with the acinar and papillary type in 86 cases (50%)”.

Question 2:  For the sake of better illustration, the radiological presentation should include figures of characteristic CTR on CT scans with corresponding histological slides. 

Answer 2: Into the material and method section at page 3 we included: Figure 1. Presentation at chest CT scan of a single pulmonary nodule: A) pure GGO (CTR 0); B) GGO with a solid component CTR ≤0.5, C) GGO with a solid component 0.5<CTR≤0.75, and D) GGO with a solid component 0.75<CTR≤1.0.

Question 3: Minor revisions

22 In Abstract CTR explain what means

33 lepidic growth of tumor cells 

65 less significant ...maybe ... smaller  or less presented

Answer 3: All minor revisions have been made in track change as suggested by the reviewer.

Besides, we improved the figures’ quality following the journal instruction and revised the English.

Reviewer 2 Report

Comments and Suggestions for Authors

This is a retrospectrive review of 172 patients who underwent mostly anatomic lung resection for adenocarcinoma of lung (either pure GCO or with some solid component). The pathological reports of resected tumors were compared with radiological findings. Main finding was that radilogically pure GCOs were lessinvasive in histology and invasiveness increased directly in propotion to the percentage of the solid component. the paper is wellwritten and informative. medhods are adequate, results well presented and conclusions justified basedon results.

Author Response

Thank very much to the reviewer for the comment.

Reviewer 3 Report

Comments and Suggestions for Authors

This single-center retrospective study evaluated 172 patients with lung adenocarcinoma presenting as ground-glass opacities (GGOs). Pure GGOs were less invasive, well-differentiated, and associated with favorable outcomes, while partial solid GGOs showed higher invasiveness and worse histological grades. Recurrence occurred in 12.2% of patients, mainly in older individuals, those with higher clinical stage, and those who underwent sublobar resections. The study concludes that GGO presence itself is a positive prognostic factor, but surgical approach and clinical stage remain critical determinants of recurrence and survival. This is an interesting topic in Oncology. However, several sections of the manuscript require revision and clarification before it can be considered for publication.

  1. Given the long study period (2006–2022), please clarify whether changes in imaging protocols, surgical techniques, or pathological classifications occurred, and how standardization across these years was ensured.
  2. As lymphadenectomy was performed in only 84% of patients, please clarify whether the absence of systematic nodal dissection may have affected staging accuracy and recurrence outcomes.
  3. With only 21 recurrence events (12.2%), please discuss whether the multivariate analyses were adequately powered and acknowledge potential statistical limitations.
  4. Although OS and CSS did not differ between pure and partial solid GGOs, relapse was higher after sublobar resections. Could this reflect selection bias, with smaller and less invasive tumors preferentially treated with sublobar resections?
  5. The manuscript compares findings with JCOG and CALGB trials, but those studies had more rigorous patient selection and randomization. How do the authors reconcile differences between their retrospective single-center results and prospective multicenter trial data?

Author Response

Question 1: Given the long study period (2006–2022), please clarify whether changes in imaging protocols, surgical techniques, or pathological classifications occurred, and how standardization across these years was ensured.

Answer 1: We added into the material and method section at page 3 all the details regarding staging, surgical technique and pathological classification.

In particular, our preoperative staging was the same since 2006 (see page 3 line 102). Also, the surgical technique and the different approaches did not change considering that we performed VATS and RATS since 2006; we added at page 3 line 118 the phrase “Indication for the different surgical approach was based only on surgeons and patients’ preference, and it did not change over the years considering that we started performing RATS and VATS in 2006”. Regarding the pathological staging we added at page 3 line 112 the phrase “Patients were all stage based on the 8th TNM AJCC staging system [14]; patients staged in the past with different TNM editions were restage based on the recent one (8th TNM ed.)”.

Question 2: As lymphadenectomy was performed in only 84% of patients, please clarify whether the absence of systematic nodal dissection may have affected staging accuracy and recurrence outcomes.

Answer 2: Lymphadenectomy details were specified already into the result section. In particular, in table 2 lymphadenectomy was included into the univariate analysis for relapse-free survival without being statistically significant, and thus not included into the multivariate analysis.

Lymph node metastases were found in only 2 (1.2%) cases; both of them were nodule with radiologic solid component (between 0.75 and 1) and invasive adenocarcinomas. However, lymphadenectomy did not have any impact on staging and survival as it showed the univariate analysis; besides, none had lymph node recurrence and only 2 (1.1%) patients died with disease.

We modified into the discussion at page 11 line 282 the paragraph “Also in our study, only 1.2% of tumors showed lymph node metastasis into the final pathological analysis, even if lymphadenectomy was not performed in all patients but in 84% of the cases. However, lymphadenectomy was not a significant prognostic factor, and none of the patients had lymph node relapses”.

Question 3: With only 21 recurrence events (12.2%), please discuss whether the multivariate analyses were adequately powered and acknowledge potential statistical limitations.

Answer 3: We agree that the total number of events (n=21) is small to perform robust multivariable analysis, but our final model includes only 3 binary variables, which is not far from the rule of thumb that Cox models should be used with a minimum of 10 outcome events per predictor variable (EPV). In fact, using simulations, Vittinghoff et al. found that confidence intervals coverage and bias were within acceptable levels despite less than 10 EPV (Vittinghoff E, McCulloch CE. Relaxing the rule of ten events per variable in logistic and Cox regression. Am J Epidemiol. 2007 Mar 15;165(6):710-8. doi: 10.1093/aje/kwk052. PMID: 17182981).

Question 4: Although OS and CSS did not differ between pure and partial solid GGOs, relapse was higher after sublobar resections. Could this reflect selection bias, with smaller and less invasive tumors preferentially treated with sublobar resections?

Answer 4: This finding is actually very much similar to the JCOG trials where recurrences were definitely higher into the sublobar group than the lobectomy one, even if survival was the same. Our study, being retrospective and with 172 patients, doesn’t have the power to demonstrate it, and besides its aim was definitely not this. However, also in our study we have higher incidence of relapses in particular into the sublobar group, pointing out the attention on sublobar resection even when perform in small nodule and with GGO pure or partial solid, which usually have less aggressive behaviour. Even if this was not the aim of our study (comparing sublobar and lobar resection) , we believe it was an important finding to show to the reader. Most of the time surgeons think that GGOs or partial solid lesion have “good behaviour” and allow the surgeon to perform sublobar resection. This finding showed that we should be aware of possible relapses.

We added into the discussion at page 12 line 319 the paragraph “Even if this was not the aim of our study (comparing sublobar and lobar resection), this finding showed that, even in case of GGOs or partial solid lesion, surgeons need to be aware of possible relapses in case of sublobar resection sublobar resection, and always rigorously follow rules and guidelines for sublobar resections”.

Question 5:  The manuscript compares findings with JCOG and CALGB trials, but those studies had more rigorous patient selection and randomization. How do the authors reconcile differences between their retrospective single-center results and prospective multicentre trial data?

Answer 5: We definitely do not want to compare our study with a randomized prospective trial, but we just used them to reflect on our results.

Round 2

Reviewer 3 Report

Comments and Suggestions for Authors

The authors have satisfactorily addressed all the questions and comments. Therefore, I believe the manuscript is suitable for publication.